# Identifying Individuals Who Currently Report Feelings of Anxiety Using Walking Gait and Quiet Balance: An Exploratory Study Using Machine Learning

**DOI:** 10.3390/s22093163

**Published:** 2022-04-20

**Authors:** Maggie Stark, Haikun Huang, Lap-Fai Yu, Rebecca Martin, Ryan McCarthy, Emily Locke, Chelsea Yager, Ahmed Ali Torad, Ahmed Mahmoud Kadry, Mostafa Ali Elwan, Matthew Lee Smith, Dylan Bradley, Ali Boolani

**Affiliations:** 1Department of Medicine, Lake Erie Osteopathic College of Medicine, Elmira, NY 14901, USA; mstark07311@med.lecom.edu; 2Department of Computer Science, George Mason University, Fairfax, VA 22030, USA; hhuang25@gmu.edu (H.H.); craigyu@gmu.edu (L.-F.Y.); 3Department of Physical Therapy, Hanover College, Hanover, IN 47243, USA; martinr@hanover.edu; 4Department of Biology, Clarkson University, Potsdam, NY 13699, USA; mccartrm@clarkson.edu; 5Department of Psychology, Clarkson University, Potsdam, NY 13699, USA; 6Department of Chemistry and Biomolecular Science, Clarkson University, Potsdam, NY 13699, USA; lockeee@clarkson.edu; 7Department of Neurology, St. Joseph’s Hospital Health Center, Syracuse, NY 13203, USA; chelsea.yager@sjhsyr.org; 8Department of Physical Therapy, Clarkson University, Potsdam, NY 13699, USA; ahmed_alimohamed@pt.kfs.edu.eg (A.A.T.); ahmed_tabia@pt.kfs.edu.eg (A.M.K.); mostafa.ali@pt.bsu.edu.eg (M.A.E.); 9Faculty of Physical Therapy, Kafrelsheik University, Kafr El Sheik 33516, Egypt; 10Faculty of Physical Therapy, Beni-Suef University, Beni-Suef 62521, Egypt; 11Department of Environmental and Occupational Health, School of Public Health, Texas A&M University, College Station, TX 77843, USA; matthew.smith@tamu.edu; 12Canino School of Engineering Technology, State University of New York, Canton, NY 13617, USA; bradley.dylan028@gmail.com

**Keywords:** anxiety, gait, mCTSIB, balance, sensors, APDM monitors, machine learning

## Abstract

Literature suggests that anxiety affects gait and balance among young adults. However, previous studies using machine learning (ML) have only used gait to identify individuals who report feeling anxious. Therefore, the purpose of this study was to identify individuals who report feeling anxious at that time using a combination of gait and quiet balance ML. Using a cross-sectional design, participants (n = 88) completed the Profile of Mood Survey-Short Form (POMS-SF) to measure current feelings of anxiety and were then asked to complete a modified Clinical Test for Sensory Interaction in Balance (mCTSIB) and a two-minute walk around a 6 m track while wearing nine APDM mobility sensors. Results from our study finds that Random Forest classifiers had the highest median accuracy rate (75%) and the five top features for identifying anxious individuals were all gait parameters (turn angles, variance in neck, lumbar rotation, lumbar movement in the sagittal plane, and arm movement). Post-hoc analyses suggest that individuals who reported feeling anxious also walked using gait patterns most similar to older individuals who are fearful of falling. Additionally, we find that individuals who are anxious also had less postural stability when they had visual input; however, these individuals had less movement during postural sway when visual input was removed.

## 1. Introduction

Mental health is a global problem, and the prevalence of anxiety ranged anywhere from 2.4% to 29.8% globally in 2013 [1]. However, since the start of the global pandemic, these numbers have increased by approximately 25.6% [2]. While we are unaware of the global toll of the pandemic on feelings of anxiety in otherwise healthy individuals, a recent study by Twenge and Joiner (2020) reported that US adults are more than three times as likely to report being anxious in 2020 than they were in 2019 [3]. In 2010, anxiety was the sixth leading cause of disability globally, with annual costs of anxiety-related disability ranging from $42 billion to $53 billion [4]. While anxiety disorders have a significant societal burden, individuals without a diagnosis of anxiety disorder may also report an increase in feelings of anxiety as a normal response to worry from a perceived stressful event [5].

Feelings of anxiety are usually self-reported, which can be influenced by an individual’s culture [1] and gender [6] differences. While there are significant advantages to using self-reported measures for anxiety, such as their high correlation with complementary objective measures, there is a risk of social and cognitive biases [7]. These biases are especially more pronounced in studies assessing mental health because of the stigma surrounding the issue. For example, Bird and Reiker [6] report that women are more comfortable expressing emotions than men, which may partially explain the high prevalence of anxiety in women compared to men [8]. Additionally, a systematic review and meta-regression by Baxter and colleagues [1] suggest that there are cultural differences in self-reported feelings of anxiety. Taken together, these findings suggest that researchers explore objective direct and indirect measures, such as biomarkers and walking gait or balance, which may be impacted by feelings of anxiety.

While significant evidence has identified objective biomarkers associated with anxiety [9], these biomarkers are individualized, which makes identifying individuals who report anxiety through biomarkers challenging, as reported by the null findings of Boeke and colleagues [10]. However, utilizing a person’s gait is one area where researchers have successfully identified feelings of anxiety through objective measures. For example, Zhao and colleagues reported a 74% accuracy rate in identifying feelings of anxiety using a walking gait [11], while Miao and colleagues [12] reported a 78.4% accuracy rate in identifying individuals who reported feeling anxious. These findings are promising because they suggest that gait may be a valid objective indirect measure to identify feelings of anxiety.

Zhao and colleagues [11] used the X-Box Kinect to capture gait data; however, they did not report gait features that were most important to identifying individuals who reported feelings of anxiety. Conversely, Miao and colleagues [12] utilized a digital camera and reported capturing features such as nose and ear movement, data that are not typically reported in the gait literature. Miao and colleagues’ [12] significant findings included associations between anxiety and joint movements of the elbow, knee, hip, and wrist, information that is also not usually reported in gait literature. The authors deliberately chose to not calculate the gait measures most commonly reported in the literature, such as stride length, walking speed, or arm swing, because they felt that time-domain data were better for predicting anxiety than traditional biomechanical variables [12]. These findings provide limited interpretability for researchers seeking to replicate these findings using traditionally measured gait characteristics. Additionally, although X-Box Kinect data are highly reliable, it produces a copious amount of data, which makes analyses more cumbersome, and may be impractical in many situations. Conversely, Inertial Movement Unit (IMU) sensors, such as the ones used in this study (APDM, Portland, OR, USA), may offer less data-intensive ways of capturing kinematic gait data. Therefore, this study aims to use the most commonly reported kinematic gait variables captured by IMU sensors to predict feelings of anxiety.

Although Zhao and colleagues [11] and Miao and colleagues [12] did not report the gait variables most commonly reported in the literature, other studies [13,14,15,16,17,18] suggest that there is a strong association between commonly measured gait characteristics and anxiety. For example, Feldman and colleagues [13] identified slower walking gait speed, shorter step length, and fewer steps per minute among adults who reported anxiety compared to those who did not. However, Feldman and colleagues [13] compared individuals who reported a diagnosis of generalized anxiety disorder compared to those who did not. These findings [13] have limited applicability to healthy individuals who may report feeling anxious without a diagnosis of generalized anxiety disorder.

Conversely, multiple studies that examined anxiety at the moment have reported changes in both gait and balance measures among healthy young and older adults [14,15,16,17,18,19]. This body of literature has reported a decrease in gait velocity with increased fear of falling [14,16,17] and differences in gait strategies when young and older adults are asked to walk under threatening or anxiety-inducing conditions [16,19]. Additionally, researchers also report that feelings of anxiety at the moment also significantly impair balance performance in healthy populations [18,19]. Taken together, we may surmise that when using traditional gait and balance measures, healthy individuals who report feeling anxious may report a slower walking gait [14,16,17] and balance dysfunction [18,19,20] compared to individuals who do not report feeling anxious.

Although there are no definitive mechanisms that link gait, balance, and feelings of anxiety, based on previously published studies, we may hypothesize a biological link exists between feelings of anxiety, gait, and balance control. Individuals who are fearful of falling utilize strategies where they rely on the spinal reflex pathway that allows for faster processing of information [21]. Further, evidence suggests that anxiety influences oculomotor and gaze control [22], which may alter locomotion in individuals who are anxious [16]. Other data suggest that individuals who report being anxious have lower attentional control but higher attentional demands for locomotion [23,24]. When examining balance control, we find that individuals who report feeling anxious have decreased visual processing [25] and increased physiological arousal, which results in a reduced center of pressure displacement and increased leg muscle activity [26,27,28]. Taken together, these findings [16,21,22,23,24,25,26,27,28], and the findings of studies that have measured changes in gait and balance due to increased feelings of anxiety [14,15,16,17,18,19], suggest that our study may be able to identify individuals who report being anxious using a combination of gait and balance parameters.

Currently, we are aware of two studies [11,12] that have successfully identified individuals who report feeling anxious from individuals who do not report feeling anxious using a walking gait. However, these studies identified gait characteristics not normally reported in the gait literature (i.e., ear movement, nose movement, and movement in the left ankle), limiting the interpretability by researchers interested in understanding traditional gait parameters needed to most accurately identify individuals who report being anxious.

Therefore, to add to the existing literature, our exploratory study aims are the following: (1) identify variables most important for machine learning algorithms to utilize to identify individuals who report feeling anxious; (2) determine the accuracy of machine learning algorithms that use clinical variables to identify individuals who report feeling anxious compared to those who do not; (3) identify differences in the various clinical measures of gait and balance between individuals who report feelings of anxiety compared to those who do not.

## 2. Methods

### 2.1. Study Design

A cross-sectional study design was used to identify individuals who reported feeling anxious in the moment compared to those who did not.

### 2.2. Participants

Participants from the community were recruited using word of mouth, flyers, campus-wide emails at the university and in-person announcements made in large classes (>30 students). To be eligible for this study all participants had to be between the ages of 18–36. Individuals were excluded from this study if they reported no neurological conditions (e.g., stroke, Parkinson’s Disease), lower-extremity orthopedic surgeries or injuries within the last 6 months, wounds of abscess on the plantar surface of their feet, uncorrectable visual impairments, and/or the inability to ambulate for 2 min without an assistive device and stand independently without pain and/or discomfort. One hundred and forty volunteers were screened and 133 were qualified to participate in this study.

### 2.3. Instruments

#### 2.3.1. Self-Reported Feelings of Anxiety

The 30-item Profile of Mood States-Short Form (POMS-SF) [29] was used to assess current feelings of anxiety. Participants reported their current intensity of subjective mood states on a 5-point Likert scale ranging from “Not at all” (scored as 0) to “Extremely” (scored as 4). Feelings of anxiety were calculated using a combination of five questions (Tense + Shaky + Nervous + Anxious + Uneasy). The Cronbach’s alpha for current feelings of anxiety was 0.839. 

#### 2.3.2. Gait

Using methodology from previous literature that used machine learning to identify moods and emotions using walking gait, this study used a 6 m × 1 m track where participants were asked to walk back and forth for two minutes at their own pace [11,30]. However, unlike previous studies that utilized the X-Box Kinect, this study utilized APDM mobility Lab^TM^ (APDM Inc, Portland, OR, USA) as they have been proven to be valid and reliable forms of gait measurements using similar protocols [31,32]. The APDM mobility Lab^TM^ sensors are a set of wireless, Opal^TM^ inertial sensors, each with a docking station that has an access point for wireless data transmission and sub-millisecond synchronization of the independent sensors. The Opal^TM^ inertial sensors have a range of 6 m and contain tri-axial accelerometers, gyroscopes, and magnetometers. The accelerometers measure linear acceleration, gyroscopes measure angular velocity, and the magnetometers measure heading with respect to the earth’s magnetic field. In this study, we used seven Opals^TM^ that were attached to the body using Velcro^TM^ straps [32]. The Opals^TM^ were placed at the following locations: lumbar region (5th lumbar vertebra), sternum (body of the sternum immediately superior to the xyphoid process), forehead (middle of the frontal bone, approximately 2.5 cm above the nasal bone), right and left foot (on the metatarsals, directly superior to the metatarsophalangeal joint), and the right and left wrist (immediately superior to the radioulnar joint) [31,33]. These locations were used because they also allowed the researchers to measure anticipatory postural adjustments prior to gait initiation [33].

#### 2.3.3. Balance

Previous literature that has examined state anxiety and balance has utilized either force plates [19] or the Neurocom Balance Master to perform the Sensory Organizational Test (SOT) [20]. These authors chose to use the modified Clinical Test of Sensory Interaction and Balance (mCTSIB) to measure balance using the Opal^TM^ monitors as mCTSIB measures have been shown to be highly correlated with scores on the SOT [34]. The mCTSIB is a clinically relevant measure of balance that can be completed using the Opal^TM^ monitors [35,36]. During the performance of this test, participants were asked to stand quietly for 30 s with feet spaced apart by the APDM footplate (a clear trapezoidal plate) and hands were placed on the hips. Balance was tested under the following 4 discrete conditions: A) Eyes open (EO) firm surface, B) Eyes closed (EC) firm surface, C) EO foam surface (Airex Balance Pad Foam Balance Board Stability Cushion, 50.5 cm W × 40.1 cm D × 6.0 cm H), and D) EC foam surface [35,36].

#### 2.3.4. Procedure

After screening for inclusion/exclusion criteria participants were scheduled for one, 75 min long session. Participants were asked to refrain from consuming alcohol, caffeine, medications, and illicit drugs at least 24 h prior to testing. Participants were invited to the lab where they completed a series of surveys on Surveymonkey.com (San Mateo, CA, USA) to determine if they had followed pre-testing instructions. Participants who reported consuming alcohol caffeine, medications, or illicit drugs in the past 24 h were re-scheduled.

Those who were eligible to complete the study were assigned a random 5-digit ID using randomizer.org and fitted for the APDM mobility monitors. They completed the POMS-SF to determine their current moods states. Participants also completed these pre-testing surveys using Surveymonkey.com using a Hewlett Packard Pavilion 15.6” Flagship Laptop (model # B018YIGHVK, Hewlett Packard, Palo Alto, CA, USA). The entire pre-testing procedure lasted approximately 10 min. After completion of the pre-test questions, participants completed the mCTSIB (approximately 2.5 min) and the two-minute walk. The mCTSIB was performed at the edge of the walking track and the two-minute walk was started immediately following the performance of the mCTSIB.

Subsequently, participants’ height was measured using a stadiometer (SECA model 220 Crothal Healthcare, Chino, CA, USA) and their weight was measured using the Tanita Body Composition Analyzer TBF-410 (Tanita Corporation, Tokyo, Japan). Participants were then asked to complete a series of surveys that asked them about health-related lifestyle factors such as diet, sleep quality, physical activity levels, and trait energy and fatigue, which are not reported in this study.

### 2.4. Statistical Analysis

#### Pre-Processing of Data

Data from SurveyMonkey.com were downloaded and exported into Microsoft Excel (Microsoft Inc., Redmond, WA, USA) where mood data were scored. Gait and balance data were exported as h5 files into Python (version 3.7, Python Software Foundation, Wilmington, DE, USA), where balance and gait characteristics for each participant were calculated. For gait measures, we calculated mean measures for each gait parameter, variation in each gait parameter by calculating standard deviation in the movement over 2 min (referred to as variation in this study). For upper and lower-extremity movement, means were calculated for each limb and means were also calculated for both limbs, further variations in inter- and intra-limb movements were also calculated. The following formulas were used:

Mean for each gait parameter (i.e., turn velocity):$$\left(mean\;turn\;velocity=\frac{sum\;of\;velocity\;of\;all\;turns}{number\;of\;turns}\right)$$

Variation for each gait parameter (i.e., standard deviation in turn velocity):$$\left(variation\;in\;turn\;velocity=\sqrt{\frac{\sum_{i=1}^{i=number\;of\;turns}{\left(turn\;velocity-mean\;turn\;velocity\right)}^{2}}{Number\;of\;turns}}\right)$$

Mean gait parameter for individual limb (i.e., mean gait speed for right leg only):$$\left(mean\;gait\;speed\;right\;leg=\frac{\;sum\;of\;gait\;speed\;of\;right\;leg\;during\;2-minute\;walk}{number\;of\;steps\;taken\;by\;right\;leg\;in\;2-minute\;walk}\right)$$

Variation in movement for each individual limb (i.e., standard deviation for right leg gait speed only/intra-limb variation in movement):$$(variation\;in\;right\;leg\;gait\;speed=\sqrt{\frac{\sum_{i=1}^{i=number\;of\;steps\;on\;right\;leg}{\left(right\;leg\;gait\;speed-mean\;right\;leg\;gait\;speed\right)}^{2}}{Number\;of\;steps\;on\;right\;leg}}$$

Mean gait parameter for both limbs of upper and lower extremity (i.e., mean gait speed):$$Mean\;gait\;speed=\frac{\sum (gait\;speed\;right\;leg+gait\;speed\;left\;leg)}{Total\;number\;of\;steps}$$

Imbalance in gait parameters between limbs (i.e., variation in gait speed between limbs/inter-limb variation in movement) to assess for consistency of movement and synchronization of movement on both sides [37,38].
$$\begin{array}{l} Imbalance\;in\;gait\;speed\;between\;limb\\ \;\;\;\;\;\;\;\;\;\;\;\;\;\;\;\;\;\;\;\;\;\;\;\;=\sum \frac{gait\;speed\;right\;leg-gait\;speed\;left\;leg}{gait\;speed\;right\;leg+gait\;speed\;left\;leg}/Total\;number\;of\;steps \end{array}$$

Mood scores were then uploaded into Python (version 3.8.5, Python Software Foundation, Wilmington, DE, USA) and merged with gait and balance data. Records were deleted if they were not collected properly or had features with missing values greater than 5%. Data were then visualized in Python. Anxiety scores were split into statistical quartiles based on frequency of distribution. The quartiles included POMS-SF Anxiety scores = 0, POMS-SF Anxiety scores = 1, POMS-SF Anxiety scores = 2, and POMS-SF Anxiety scores ≥ 3. For the purposes of this exploratory study, the researchers were only interested in determining whether the models could classify individuals on the extreme ends. Therefore, this study only included individuals who reported a POMS-SF Anxiety scores = 0 and those who reported scores ≥ 3 in the final analysis. Therefore, the researchers grouped participants by those who reported scores of ≥3 on the POMS-SF anxiety (Anxious) and those who reported POMS-SF anxiety score of 0 (Not Anxious). Individuals who reported a score of 1 or 2 were not included in our final analysis. After pre-processing data, we had data for 88 participants with 190 gait characteristics, and 30 features for each of the four balance conditions (i.e., 120 total features for balance) used in the analyses. The researchers then filled in missing values through the mean value of that feature in the dataset [35]. Chi-square goodness of fit was used to determine sex differences and independent sample *t*-tests were used to determine differences in age, height, and weight between the two groups.

### 2.5. Main Analysis

#### 2.5.1. Objective 1: Feature Importance

When recording high dimensional features, not every feature is equally important, and there may be many redundant features that are of less importance. Therefore, to sort through the 310 features and 71 valid data points collected during this study, the researchers used the Random Forest (RF) according to their feature importance [39].

#### 2.5.2. Objective 2: Model Training

After sorting the features, the dataset was used to train the model through classifiers. The researchers classified the records who reported scores ≥ 3 on the anxiety portion of the POMS-SF as Anxious and those who scored a 0 as Not Anxious. The researchers used all features and top five features (using 0.024 as a cut-off for feature importance) to train each model. The researchers trained the models in a 10-fold cross-validation manner to avoid problems such as overfitting or selection bias to some degree [40]. The data were randomly split into the training set (90%) and test set (10% and ran each of the ML models 10,000 times using a Monte Carlo method [41,42,43]. Further information may be found in Appendix A.

#### 2.5.3. Objective 3: Mean Differences

An analysis of co-variance was used to determine statistically significant differences between individuals who reported feeling anxious compared to those who did not while accounting for sex, age, height, and weight.

## 3. Results

There were 51 individuals (21 males and 30 females) who reported not feeling anxious, while 36 individuals (14 males and 22 females) who reported feeling anxious (*p* = 0.830). There was no significant difference (*p* > 0.05) in height or weight between individuals who reported feeling anxious (height = 174.09 ± 8.10 cm, weight = 74.86 ± 14.26 kg) compared to those who reported not feeling anxious (height = 174.13 ± 9.76 cm, weight = 73.00 ± 15.83 kg). There was a statistically significant difference in age between the two groups (*p* = 0.034), with the anxious group reporting being significantly younger (23.31 ± 3.69 years) than the non-anxious group (25.20 ± 4.24 years).

### 3.1. Objective 1: Feature Importance

The top five features using an RF were the mean angles of turns, the variance of neck bending in the frontal plane, variance in arm swing speed, movement of the lumbar region in the sagittal plane, and the maximal lumbar rotation in the transverse plane (Appendix A).

### 3.2. Objective 2: Model Training

Our top model was a RF with the top five features. The median accuracy was 75%, and the mean accuracy was 69.7 ± 16.4%. The Alpha-Beta (AB) using the top five features was the next most accurate model, with the median accuracy being 63% and the mean accuracy being 67.9%.

### 3.3. Objective 3: Mean Differences

#### 3.3.1. Gait

There were several significant differences in gait characteristics between the two groups. Full results may be found in Appendix A. Significant results are reported in Table 1.

#### 3.3.2. Neck Features

None of the features in the neck were in the top five selected features by our machine learning models. Outside of the top five features, individuals who reported feeling anxious also had less variance with neck rotation (2.78° ± 1.74 vs. 3.80° ± 2.95) and less neck movement along the sagittal plane (7.88° ± 2.67 vs. 8.96° ± 2.96) while displaying increased variance in neck rotation on the right side along the transverse plane (12.78 ± 17.13 vs. 9.55 ± 5.61), and increased variance in neck rotation to the left along the transverse plane (12.90° ± 17.14 vs. 9.63° ± 5.82).

#### 3.3.3. Trunk Features

Of our top five features, those who reported feeling anxious had a significantly greater movement of their lumbar in the sagittal plane (5.46° ± 5.10) compared to those who did not report feeling anxious (3.27° ± 4.42). For features outside of the top five, those who reported feeling anxious had significantly greater left lumbar frontal plane bend (6.94 ± 2.74 vs. 5.62 ± 2.46), greater minimum lumbar flexion/extension angle (−0.66° ± 5.02 vs. −2.96° ± 4.52), greater maximum lumbar rotation to the left (4.76° ± 12.43 vs. 0.03° ± 15.23), greater lumbar range of motion in the frontal plane (6.54° ± 2.79 vs. 5.72° ± 1.86), while having less variance in the sagittal range of motion in the trunk (thoracic spine area) (1.06° ± 0.35 vs. 1.19° ± 0.43), and less variance in trunk motion in the transverse plane (1.81° ± 0.58 vs. 2.05° ± 1.18).

#### 3.3.4. Lower Extremity Characteristics

None of the lower extremity features were in the top five; however, many of the features out of the top five were significantly different. Those who reported feeling anxious also had a significantly lower variance for gait speed between limbs (0.91% ± 0.80 vs. 1.23% ± 0.73), less step variability between legs (8.16% ± 5.85 vs. 11.21% ± 7.44), and less variance in mid-swing elevation between legs (13.88% ± 11.97 vs. 18.8% ± 13.94), and less time spent in single leg support (39.59 s ± 1.50 vs. 40.10 s ± 1.45). Self-reported feelings of anxiety also increased variance in double leg support time (0.84 ± 0.33 vs. 0.76 ± 0.15), greater variance in toe-out angle between legs (2.78% ± 1.97 vs 2.21% ± 1.70), and increased time spent in double leg support time (10.58 s± 1.34 vs 10.01 s ± 1.41).

#### 3.3.5. Turning Features

Of the top five features in our model, those who reported feeling anxious took significantly wider turns (188.30° ± 4.17) compared to those who were not anxious (185.87° ± 3.43). For features outside of the top five, those who reported feeling anxious had significantly greater variance in turn angles (5.92° ± 1.88 vs. 5.17° ± 1.29), faster turns (2.17 s ± 0.19 vs. 2.23 s ± 0.18), and less variation in the amount of time spent turning (0.21 ± 0.09 vs. 0.24 ± 0.08).

#### 3.3.6. Anticipatory Postural Adjustment during Initiation of Gait

There were no significant differences (*p* > 0.05) in any of the variables we measured that were associated with the initiation of walking gait.

#### 3.3.7. Balance

None of the balance variables were considered to be part of the top five features necessary to predict feelings of anxiety. However, several characteristics of balance were significantly different between individuals who reported feeling anxious compared to those who did not. The full set of results may be found in Appendix A. Significant results are only reported in Table 2, Table 3, Table 4 and Table 5.

#### 3.3.8. Condition: Eyes Open, Feet on Ground

Individuals who reported feeling anxious reported a faster postural sway velocity (0.19 m/s ± 0.26 vs. 0.12 m/s ± 0.10), faster velocity during postural corrections in the sagittal plane (0.18 m/s ± 0.26 vs. 0.11 m/s ± 0.08), larger postural sway area (1.24° ± 2.31 vs. 0.86° ± 0.93), larger root mean square (RMS) sway (0.52° ± 0.62 vs. 0.37° ± 0.20), greater acceleration in the y-axis (0.21 m/s^2^ ± 0.25 vs. 0.15 m/s^2^ ± 0.08), larger ellipses radius in the y-axis (1.25° ± 1.52 vs. 0.87 ± 0.49), greater RMS sway in the sagittal plane (0.51° ± 0.62 vs. 0.34° ± 0.18), larger ellipsis sway area (0.04 m/s^4^ ± 0.07 vs. 0.02 m/s^4^ ± 0.03), greater postural sway acceleration range (0.47 m/s^2^ ±0.63 vs. 0.35 m/s^2^ ± 0.26), greater postural acceleration range in the sagittal plane (0.45 m/s^2^ ± 0.63 vs. 0.32 m/s^2^ ± 0.25), and greater RMS sway in the sagittal plane (0.09 m/s^2^ ± 0.10 vs. 0.06 m/s^2^ ± 0.03). Self-report of anxiety also produced less jerk in the coronal plane (0.41 m^2^/s^5^ ± 0.34 vs. 0.57m^2^/s^5^ ± 0.82), less acceleration range in the coronal plane (0.11 m/s^2^ ± 0.06 vs. 0.13 m/s^2^ ± 0.09), less postural sway acceleration in the coronal plane (0.02 m/s^2^ ±0.01 vs. 0.02 m/s^2^ ± 0.02), greater postural sway RMS sway (0.09 m/s^2^ ± 0.10 vs. 0.06 m/s^2^ ± 0.03), and lower RMS sway angle in the coronal plane (0.11° ± 0.05 vs. 0.13° ± 0.11) (Table 2).

#### 3.3.9. Condition: Eyes Closed, Feet on Ground

Individuals who reported feeling anxious had smaller ellipses during postural rotations (1.55 m^2^ ± 0.14 vs. 1.62 m^2^ ± 0.25), less jerk in the coronal plane (0.37 m^2^/s^5^ ± 0.28 vs. 0.54 m^2^/s^5^ ± 0.67), and smaller postural sway area (1.55°^2^ ± 0.14 vs. 1.62°^2^ ± 0.25) (Table 3).

#### 3.3.10. Condition: Eyes Open, Feet on Foam Surface

Individuals who reported feeling anxious had a slower postural sway acceleration range in the frontal plane (0.17 m/s^2^ ± 0.05 vs. 0.21 m/s^2^ ± 0.07), less postural sway frequency dispersion (0.66 ± 0.06 vs. 0.68 ± 0.04), smaller ellipses along the x-axis (0.07 m^2^ ± 0.02 vs. 0.08 m^2^ ± 0.03), lower root mean square in the coronal plane (0.18° ± 0.05 vs. 0.21° ± 0.07), less frequency dispersion in the sagittal plane (0.69 ± 0.04 vs. 0.71 ± 0.04), lower RMS sway in the coronal plane (0.03 m/s^2^ ± 0.01 vs. 0.04 m/s^2^ ± 0.01), smaller ellipses sway area (0.04 m/s^4^ ± 0.02 vs. 0.05 m/s^4^ ± 0.03), less jerk in the coronal (0.78 m^2^/s^5^ ± 0.47 vs. 0.99 m^2^/s^5^ ± 0.82), and sagittal (1.56 m^2^/s^5^ ± 0.88 vs. 1.92 m^2^/s^5^ ± 1.75) planes and larger centroidal frequency in the coronal plane (1.09 Hz ± 0.15 vs. 1.00 Hz ± 0.24) (Table 4). 

#### 3.3.11. Condition: Eyes Closed, Feet on Foam Surface

Individuals who reported feeling anxious had a slower acceleration in the ellipsoid motion (1.51 m/s^2^ ± 0.30 vs. 1.63 m/s^2^ ± 0.25), less postural frequency dispersion in the coronal plane (0.63 ± 0.07 vs. 0.65 ± 0.05), less sway area rotation (1.51° ± 0.30 vs. 1.63° ± 0.25), and less jerk in the coronal plane (2.09 m^2^/s^5^ ± 1.31 vs. 2.93m^2^/s^5^ ± 3.24) (Table 5).

## 4. Discussion

To the knowledge of the researchers, this is the first study to utilize sensors and machine learning to identify current feelings of anxiety using gait and balance measures. Although the accuracy of the models in this study is aligned with previously reported literature that identified feelings of anxiety over the last 2 weeks [11,12], the findings of this study add significantly to the literature by reporting gait characteristics that have clinical meaning and measures that can be used to identify individuals who currently report being anxious. Although significant evidence indicates that there are differences in gait [14,15,16,17,18,19] and balance [18,19] among individuals who report feeling anxious compared to those who do not, the findings of this study suggest that gait may be the most important feature to consider when using sensors to identify individuals who are anxious in the moment.

### 4.1. Objective 1

The findings from these analyses suggest that the turning angle, mean lumbar movement, and variations in the neck and arm movements are the most important features in predicting anxiety. To the knowledge of the researchers, there is limited literature to which these findings can be compared. When examining the literature, findings report that older adults usually adjust their turns [44], as turning requires maintaining balance [45] when they report feeling anxious. These findings are similar to the findings of this study, as young adults who reported feeling anxious in this study made wider turns compared to those who were not anxious. This finding is unique in that the previous literature that has used machine learning in walking gait to identify feelings of anxiety has used variations in movements (i.e., nose and ear movement, ankle movement) [11,12], whereas this study provides the gait characteristics most commonly used in the literature.

### 4.2. Objective 2

The median of the best model had a 75% accuracy, which is congruent with previously reported literature [11,12]. However, the findings of this study add the following three unique aspects to the literature: (1) This study used sensors, which are significantly less data-intensive than the X-Box Kinect; (2) this study was able to identify individuals who reported feeling anxious in the present, while previous literature identified individuals who reported feeling anxious at any point over the last two weeks; (3) this study used measures most often reported in gait literature to identify feelings of anxiety, while the previous literature that reported variables not often reported (i.e., nose and ear movement).

### 4.3. Objective 3

#### 4.3.1. Gait

Overall, our findings suggest that individuals who report feeling anxious had no trouble initiating gait; however, based on the variables that were significantly different, these individuals had more cautious gait patterns. The gait patterns found in the anxious young adults in our study are similar to those observed in older adults who are fearful of falling [14]. Individuals who self-report being anxious show a significantly greater side-to-side rotation of the neck, but less up-and-down movement. These findings may be explained by the fact that anxiety has an effect on oculomotor and gaze control [22], which has been linked to alterations in locomotion [16]. The findings of this study suggest a reduced smoothness of neck movement among individuals who report feeling anxious, which may be explained by previous hypotheses that suggest that individuals who report feeling anxious are consistently performing threat assessments during locomotion [16].

When examining movements of the trunk and lumbar spine, our results found that anxious individuals had an increased forward/backward and side-to-side movement as they walked compared to individuals who did not report feeling anxious. These findings also report that anxious individuals had a significantly greater bending to the left side when compared to healthy individuals, but not on the right side. Our findings may be explained by the fact that, in this study, participants were asked to walk around an oval track making left turns only. The increased bending to the left by anxious individuals suggests that these individuals were trying to create a base of support when turning. While this study reports increased variations in lumbar movements, the results also found decreased variance in thoracic movements. Taken together, this suggests that individuals who are anxious may be trying to use the spinal reflex pathway (automatic control strategy) to control locomotion rather than send information to the higher centers (executive strategy). Utilizing this strategy allows for faster unconscious correction of the trunk (i.e., fast, parallel processing of information) and is utilized by individuals who are fearful of falling [21].

The lower extremity movements of anxious young adults are also similar to those of older adults who are fearful of falling [46]. For example, this study reports that there were fewer variations in the lower extremity movements among anxious individuals and that anxious individuals also spent more time in double leg support time, thus less time in single leg support time when compared to their counterparts who reported not feeling anxious. The lower extremity movements further reinforce the findings presented for trunk and lumbar movements because they suggest a lower attentional control, but higher attentional demands for locomotion [23] among anxious individuals. Previously, a meta-analysis reported a significant negative relationship between anxiety and attentional control [24], which the authors believe may have impacted the walking gait of anxious individuals in this study. Based on the movement of the neck (increased visual scanning), trunk, and lumbar regions (decreased trunk movement), the authors hypothesize that anxious individuals may be scanning for threats by minimizing attentional control of locomotion.

Another potential hypothesis to explain walking gait differences may be that anxious individuals may also have balance deficiencies in walking gait. This hypothesis is supported by the findings in this study that show participants who reported feeling anxious had a decreased stride velocity and spent more time in double leg support, yet they had a greater variance in double leg support time, which suggests that they were trying to unsuccessfully regulate and slow gait speed over the course of the two minutes to maintain balance. This hypothesis is further supported by anxious individuals’ having an increased variance in toe-out angle and mid-swing elevation, which suggests that these individuals were trying to correct for information received by their balance system [47,48]. Additionally, this hypothesis is further supported when examining how anxious individuals turn. In these data, the authors report that anxious individuals take wider, more variable, yet faster turns. Taken together, this suggests that anxious individuals may be trying to find a larger base of support when turning (wider turns), while decreasing the amount of time spent in unstable environments (faster turns, as turns, are less stable than straight walking). These findings are similar to those reported among individuals with gait disorders who try to avoid falling [49], as well as individuals who report feeling low energy [50,51]. A visual representation of the gait differences can be found at https://gaitsim.dmanserver.com/AnxietyYN (accessed on 12 January 2022).

#### 4.3.2. Balance

Although none of the balance-associated variables were predictive of feelings of anxiety in the machine learning model presented in this study, it was found that anxious individuals had different balance corrections and errors compared to those who did not feel anxious. This study reported that anxious individuals had greater front-to-back (sagittal plane) movement; however, they had less side-to-side movement when their eyes were open and their feet were on a firm surface. The authors also found that anxious individuals correct significantly faster front-to-back but slower side-to-side compared to non-anxious individuals. Interestingly, when examining the balance condition where somatosensory input is unreliable but visual input is present, postural sway is “smoother” with less spatial displacement and slower temporal adjustments, except when correcting side-to-side. The authors also found that when vision was occluded on both firm surface and foam surface conditions, individuals who reported being anxious had a “smoother” balance with less spatial displacement and slower temporal corrections. Taken together, these findings suggest that visual input is impacted by feelings of anxiety [25], as shown by quicker temporal corrections in both vision conditions, albeit in different planes. The balance findings from this study further support the hypothesis put forth by the authors when explaining the gait findings, as balance findings suggest that individuals who are anxious may be visually scanning for threats. This hypothesis may explain greater front-back movements with quicker corrections in the eyes open, feet on firm surface conditions and quicker corrections in the eyes open, unstable surface conditions. However, these findings do not explain the smaller spatial movements in unstable environments. Smaller spatial movements in unstable environments and vision-occluded balance may be best explained by the fact that anxiety increases physiological arousal, which is associated with a reduced center of pressure displacement and increased leg muscle activation [26,27,28], although this study did not measure leg muscle activity.

#### 4.3.3. Implications

The findings of this study have implications in multiple fields, as it gives insight into how best to recognize feelings of anxiety among young adults while also providing the gait variables most often reported in the literature. The findings from this study suggest that there are signals, specifically in locomotion, which can be helpful to identify individuals who are feeling anxious using IMU sensor technology. Based on the accuracy of the results of this study, the authors suggest future researchers try to capture additional data during locomotion and should focus their efforts on trying to capture signals that document variance in locomotion. Additionally, these findings suggest that even though the participants of this study were young, those who reported feeling anxious mimicked the gait patterns most often observed among older adults who are fearful of falling. Although this study did not measure locomotion in athletic competition, these findings may be of importance to tactical and sports athlete trainers as they may find individuals who report feeling anxious are constantly threat scanning and walking in a protective manner, which may influence their athletic performance. These findings may also be of interest to clinicians working in the movement analysis realm (i.e., physiotherapists), as some of the instability/declines in balance that they may note may be due to anxiety instead of functional deficits.

#### 4.3.4. Limitations

This study is not without limitations. The primary limitation of this study is its cross-sectional design. However, a recent investigation suggests that not all individuals report feeling anxious even when using anxiety-provoking conditions [52]; therefore, suggesting that researchers trying to identify when an individual is feeling anxious may have difficulty inducing anxiety. These inter-individual differences in responses to various anxiety-provoking conditions suggest that for an exploratory study, a cross-sectional design may be ideal. Another limitation is that feelings of anxiety are self-reported measures that may have inherent reporting bias. The Hawthorne effect may also be at play here, as individuals who normally would not report feeling anxious may have been anxious when they reported to the lab [53]. The Hawthorne effect further underscores the importance of this study, as there is a need for technology that may be used to identify feelings of anxiety in the moment. Another potential limitation of the findings is that the individuals identified in this study may have had gait deviations due to factors other than anxiety that may not have been accounted for in this study (i.e., pain). Additionally, this study eliminated individuals who reported some anxiety (POMS-SF scores of 1 and 2), which may have provided additional insight into how locomotion changes as feelings of anxiety increase. Lastly, due to the small sample size, the models reported in this study had a median accuracy of 75%, and some of the best models had 100% accuracy. The models with 100% accuracy rates may have just been “lucky” in the training and test sets of data that were used. However, those models that reported 100% accuracy do provide insight into the fact that the accuracy of the models may significantly improve with larger sample sizes. Future research should examine other movements and try to perform these studies with larger sample sizes. Future studies should also try to understand the chronicity of anxiety in the population and whether individuals who report feeling anxious more frequently and/or more intensely may more permanently alter their gait patterns and/or create compensatory mechanisms in walking gait.

## 5. Conclusions

The purpose of this exploratory study was to use machine learning to identify young adults who currently report feeling anxious. The findings from this study are aligned with previous literature in that these models were able to identify individuals who reported feeling anxious with 75% accuracy. These results identified that locomotion was more important than quiet balance at identifying anxious individuals. Additionally, this study found that variances in the upper extremity, trunk, and neck movement may be the most important features in identifying anxious individuals. These findings also suggest that young adults who reported feeling anxious exhibit walking patterns similar to older adults who report fear of falling. This study also reported that quiet balance among young adults who currently report feeling anxious was different compared to those not feeling anxious. Future studies should try to capture locomotion data in larger populations while also trying to capture other signals that may help create more accurate models.

## Figures and Tables

**Table 1 sensors-22-03163-t001:** Top 5 Variables and Significant Gait Variables Only.

			Anxious		Not Anxious		
Variable	Relative Importance	Ranking	Mean	SD	Mean	SD	Significant Difference
Mean turns angle (°)	0.05	1	188.30	4.17	185.87	3.43	Yes
Variance neck bending in frontal plane (°)	0.03	2	1.74	1.18	2.04	1.14	
Variance in L arm swing velocity (°/s)	0.03	3	45.21	45.87	53.35	37.50	
Mean lumbar max. in sagittal plane (°)	0.02	4	5.46	5.10	3.27	4.42	Yes
Mean lumbar R rotation max. (°)	0.02	5	6.91	13.07	10.88	15.11	
Variance gait speed between legs (%)	0.02	7	0.91	0.80	1.23	0.74	Yes
Mean lumbar L bending max. in the frontal plane (°)	0.02	9	6.94	2.74	5.62	2.46	Yes
Variance step variability between legs (%)	0.02	12	8.16	5.85	11.21	7.44	Yes
Mean lower limb stance GCT (s)	0.01	15	60.52	1.51	59.94	1.48	Yes
Variance mid-swing elevation between legs (%)	0.01	17	13.88	11.97	18.98	13.94	Yes
Variance neck in the sagittal plane range (°)	0.01	18	3.31	2.01	3.77	1.37	Yes
Mean lumbar in the sagittal plane min. (°)	0.01	20	-0.66	5.02	-2.96	4.52	Yes
Mean lumbar L max. rot. (°)	0.01	29	4.76	12.43	0.03	15.23	Yes
Variance neck rot. range in frontal plane (°)	0.01	41	2.78	1.74	3.80	2.95	Yes
Mean lumbar bending range in frontal plane (°)	0.01	42	9.82	3.12	8.78	3.16	Yes
Variance R lower limb terminal double support (% GCT)	0.01	44	0.84	0.33	0.76	0.15	Yes
Variance toe out angle between legs (%)	0.01	45	2.78	1.97	2.21	1.70	Yes
Variance neck R max. rot. (°)	0.01	46	12.78	17.13	9.55	5.61	Yes
Mean turns duration (#)	0.00	58	2.17	0.19	2.23	0.18	Yes
Mean lumbar coronal ROM (°)	0.00	82	6.54	2.79	5.72	1.86	Yes
Variance turns angle (°)	0.00	92	5.92	1.88	5.17	1.29	Yes
Mean neck R L rotation range (°)	0.00	137	7.39	2.46	8.25	2.44	Yes
Mean R leg swing (% GCT)	0.00	149	39.48	1.51	40.06	1.48	Yes
Mean L Leg single limb support (% GCT)	0.00	179	39.59	1.50	40.10	1.45	Yes
Variance in turn duration (s)	0.00	189	0.21	0.09	0.24	0.08	Yes
Mean neck range in the sagittal plane (°)	0.00	205	7.88	2.67	8.96	2.96	Yes
Variance in neck L max. rot. (°)	0.00	243	12.90	17.14	9.63	5.82	Yes
Mean R leg terminal double support (% GCT))	0.00	257	10.58	1.34	10.01	1.41	Yes
Variance in trunk ROM in sagittal plane (°)	0.00	260	1.06	0.35	1.19	0.43	Yes
Variance in trunk ROM in transverse plane (°)	0.00	261	1.81	0.58	2.05	1.18	Yes

L = left, R = right, rot = rotation, ROM = range of motion, max. = maximum, min. = minimum, avg. = average, GCT = gait cycle time, # = number, % = percentage, cm= centimeter, GCD = gait cycle duration, ° = degrees.

**Table 2 sensors-22-03163-t002:** Significant Variables Only for Eyes Open, Feet on Ground.

			Anxious		Not Anxious		
Variable	Relative Importance	Ranking	Mean	SD	Mean	SD	Significant Difference
Mean velocity in sagittal plane (m/s)	0.01	39	0.18	0.26	0.11	0.08	Yes
Sway angle area (°)	0.00	95	1.24	2.31	0.86	0.93	Yes
RMS sway angle (°)	0.00	100	0.52	0.62	0.37	0.20	Yes
Mean velocity (m/s)	0.00	141	0.19	0.26	0.12	0.10	Yes
Jerk in coronal plane (m^2^/s^5^)	0.00	144	0.41	0.34	0.57	0.82	Yes
Acceleration 95% ellipse radius on y-axis (m/s^2^)	0.00	153	0.21	0.25	0.15	0.08	Yes
Angle 95% ellipse radius on y-axis (°)	0.00	188	1.25	1.52	0.87	0.49	Yes
Angle RMS Sway in sagittal plane (°)	0.00	217	0.51	0.62	0.34	0.18	Yes
Acceleration 95% ellipse sway area (m^2^/s^4^)	0.00	265	0.04	0.07	0.02	0.03	Yes
Acceleration range in coronal plane (m/s^2^)	0.00	283	0.11	0.06	0.13	0.09	Yes
Acceleration range (m/s^2^)	0.00	284	0.47	0.63	0.35	0.26	Yes
Acceleration range in sagittal plane (m/s^2^)	0.00	288	0.45	0.63	0.32	0.25	Yes
Acceleration RMS sway in coronal plane (m/s^2^)	0.00	292	0.02	0.01	0.02	0.02	Yes
Acceleration RMS sway (m/s^2^)	0.00	293	0.09	0.10	0.06	0.03	Yes
Acceleration RMS sway in the sagittal plane (m/s^2^)	0.00	295	0.09	0.10	0.06	0.03	Yes
Angle Durations (s)	0.00	300	29.99	0.00	29.99	0.00	Yes
Angle RMS sway in coronal plane (°)	0.00	302	0.11	0.05	0.13	0.11	Yes

rot = rotation, RMS = Root mean square, ° = degrees.

**Table 3 sensors-22-03163-t003:** Significant variables only for Eyes Closed, Feet on Ground.

			Anxious		Not Anxious		
Variable	Relative Importance	Ranking	Mean	SD	Mean	SD	Significant Difference
Acceleration 95% ellipse rot (m/s^2^)	0.01	32	1.55	0.14	1.62	0.25	Yes
Jerk in coronal plane (m^2^/s^5^)	0.00	148	0.37	0.28	0.54	0.67	Yes
Sway area rot (°)	0.00	165	1.55	0.14	1.62	0.25	Yes
Mean velocity in sagittal plane (m/s)	0.00	172	0.10	0.05	0.12	0.05	Yes
Angle durations (s)	0.00	174	29.99	0.00	29.99	0.00	Yes

rot = rotation, RMS = Root mean square, ° = degrees.

**Table 4 sensors-22-03163-t004:** Significant Variables Only for Eyes Open, Feet on Foam Surface.

			Anxious		Not Anxious		
Variable	Relative Importance	Ranking	Mean	SD	Mean	SD	Significant Difference
Velocity range in coronal plane (m/s^2^)	0.02	8	0.17	0.05	0.21	0.07	Yes
Frequency dispersion	0.01	16	0.66	0.06	0.68	0.04	Yes
Acceleration 95% ellipse radius on x-axis (m/s^2^)	0.01	19	0.07	0.02	0.08	0.03	Yes
Centroidal frequency in coronal plane (Hz)	0.01	36	1.09	0.15	1.00	0.24	Yes
Sway angle area radius in coronal plane (°)	0.01	40	0.40	0.10	0.49	0.15	Yes
RMS sway angle in coronal plane °	0.01	43	0.18	0.05	0.21	0.07	Yes
Frequency dispersion in sagittal plane	0.00	154	0.69	0.04	0.71	0.04	Yes
Acceleration RMS sway in coronal plane (m/s^2^)	0.00	176	0.03	0.01	0.04	0.01	Yes
Acceleration 95% ellipse sway area (m^2^/s^4^)	0.00	267	0.04	0.02	0.05	0.03	Yes
Jerk in coronal plane (m^2^/s^5^)	0.00	271	0.78	0.47	0.99	0.82	Yes
Jerk in sagittal plane (m^2^/s^5^)	0.00	272	1.56	0.88	1.92	1.75	Yes
Acceleration range (m/s^2^)	0.00	286	0.39	0.12	0.41	0.13	Yes
Angle 95% ellipse radius in y-axis (°)	0.00	298	0.97	0.38	0.98	0.33	Yes
Angle RMS sway in sagittal plane (°)	0.00	305	0.39	0.16	0.39	0.14	Yes
Angle sway area (°^2^)	0.00	306	1.27	0.64	1.59	0.95	Yes

rot = rotation, RMS = Root mean square, ° = degrees, °^2^ = degrees^2^.

**Table 5 sensors-22-03163-t005:** Significant Variables Only for Eyes Closed, Feet on Foam Surface.

			Anxious		Not Anxious		
Variable	Relative Importance	Ranking	Mean	SD	Mean	SD	Significant Difference
Acceleration 95% ellipse rot (m/s^2^)	0.02	13	1.51	0.30	1.63	0.25	Yes
Frequency dispersion in coronal plane	0.00	122	0.63	0.07	0.65	0.05	Yes
Rot sway area (°)	0.00	123	1.51	0.30	1.63	0.25	Yes
Jerk in coronal plane (m^2^/s^5^)	0.00	140	2.09	1.31	2.93	3.24	Yes
Angle durations	0.00	230	29.99	0.00	29.99	0.00	Yes
Acceleration RMS sway in sagittal plane (m/s^2^)	0.00	296	0.10	0.03	0.11	0.04	Yes
Angle 95% ellipse radius in y-axis (°)	0.00	299	1.49	0.45	1.57	0.58	Yes
Angle RMS sway (°)	0.00	304	0.69	0.20	0.72	0.24	Yes
Angle sway area (°^2^)	0.00	307	3.77	2.09	4.15	2.56	Yes
Jerk in coronal plane (m^2^/s^5^)	0.00	271	0.78	0.47	0.99	0.82	Yes
Jerk in sagittal plane (m^2^/s^5^)	0.00	272	1.56	0.88	1.92	1.75	Yes
Acceleration range (m/s^2^)	0.00	286	0.39	0.12	0.41	0.13	Yes
Angle 95% ellipse radius in y-axis (°)	0.00	298	0.97	0.38	0.98	0.33	Yes
Angle RMS sway in sagittal plane (°)	0.00	305	0.39	0.16	0.39	0.14	Yes
Angle sway area (°^2^)	0.00	306	1.27	0.64	1.59	0.95	Yes

rot = rotation, RMS = Root mean square, ° = degrees, °^2^ = degrees^2^.

## Data Availability

Data is available on request due to Institutional Review Board (IRB) restrictions.

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
