# Peer review of "Identifying Individuals Who Currently Report Feelings of Anxiety Using Walking Gait and Quiet Balance: An Exploratory Study Using Machine Learning"

_sensors, 2022, doi:10.3390/s22093163_

Round 1

Reviewer 1 Report

Having read the article, I think that overall the authors have done a good job. However, I think that the authors should be careful about mistakes in the text (see for example, 'rnage' on line 112). Also, in the results section, if the authors could present figures it would help the readers to understand better. 

Author Response

Dear Reviewer,

We would like to thank the reviewer for their time and excellent feedback. We feel that based on the reviewer’s feedback, our manuscript is significantly improved.

Reviewer 1

Having read the article, I think that overall the authors have done a good job. However, I think that the authors should be careful about mistakes in the text (see for example, 'rnage' on line 112). Also, in the results section, if the authors could present figures it would help the readers to understand better. 

Response- We appreciate the reviewer’s feedback. We have now proof read our manuscript and we believe we have addressed all the mistakes. Regarding the reviewer’s feedback about figures, we now present tables in our results.

Reviewer 2 Report

This is a review of the manuscript “Identifying individuals who currently report feelings of anxiety using walking gait and quiet balance: An exploratory study using machine learning”, submitted to the “Intelligent Sensors” section, special issue “Analysis of Biomedical Signals and Physical Behavior Sensing in the Development of Systems for Monitoring, Training, Controlling, and Improving Quality of Life”.

This manuscript describes a pilot study using machine learning (ML) to simultaneously monitor individuals’ gait and quiet balance to potentially identify individuals under the influence of anxiety. Although the concept may be interesting to the readers of Sensors and this special issue, especially among those clinicians who are looking for an objective and reliable surveillance and/or screening tool for psychological assessment on anxiety. However, the current manuscript was not written with clear logic and hard to follow with the presence of spelling errors, grammar issues, and redundant wordings. It could use some extensive editing work to make the main text fluent and easier to understand. More importantly, after reading the manuscript, the reviewer developed some concerns over the quality of this manuscript in its current format, particularly the study design, the way data were analyzed, and the interpretation of results. Some additional explanation would be good.

In general, based on the overall impression of the current exploratory study, the reviewer would say that the authors aimed to establish an objective approach to identify/predict feelings of anxiety using machine learning with a pool of candidate variables of the gait and balance features, as stated in the Introduction section, “to identify whether balance control, gait, or a combination of the two would be most accurate to predict if a healthy individual is feeling anxious in the moment”. To the reviewer’s knowledge, gait parameters, balance control, etc. are closely related to neuromuscular functions, which can be affected by the psychological status and/or disturbance, not necessarily limited to or caused by anxiety alone. Therefore, it appeared to be a long leap trying to establish a direct connection using gait and balance as the outcome measure of anxiety. The current Introduction section is not focused on providing the foundation work, such as the existing evidence of the associations between gait parameters and postural control capability (i.e., balance test) and the psychological features (i.e., anxiety), especially some concrete evidence explaining the biological plausibility, in what particular way anxiety would actually influence how an individual walk or stand. If the logic is not offered or clearly presented, the whole study might become a pursue of pure statistics, just throwing a whole bunch of variables out there and let the algorithm run and see what kind of relationship will show up. In the Discussion section, the authors do provide some insight when discussing Objective 1 and 3. The reviewer would suggest adding similar material in the Introduction. Most importantly, the authors would need to explain the causal relationship that may warrant a predictive relationship between how people feel and how they walk or balance. It is one thing that they are statistically associated and another thing if you can rely on one to predict another. In practice, if indeed an individual is exhibiting large deviation from the “normal” walking gait, we do not immediately seek explanations or diagnosis from a psychology perspective. Instead, clinicians will more likely look for neuromuscular and/or skeletal issues first. Therefore, what is the relevance of these findings to clinical practices?

Secondly, as the authors mentioned in the Limitations section that this is a cross-sectional design with self-reported anxiety status collected from a relatively small sample of young subjects, who were separated into two groups, one with anxiety (≥3) and the other without anxiety (=0), the current methodology may not just have limitations but flaws in logic. The current comparisons were made between datasets collected from different subjects rather than from the same subject at various status (i.e., use his/herself as reference for what is normal and what is abnormal). Unfortunately, with the vast variations in human neuromuscular-skeletal system, it is not an easy job to establish a standard that is true and applicable to all individuals, not to mention the influence of other variables, including gender, age, body composition, general health, and medical conditions. The current sample while being young and healthy, is also a substantial limiting factor on the possible application of the current findings among other samples, where subjects do have this or that condition, such as obesity, sleep problems, degenerative changes to the neuro-musculoskeletal system. With everything blended in together, how can we single out the influence of anxiety? As mentioned by the authors, since anxiety-inducing conditions can be simulated (16,19), the current study may consider establishing baseline and anxiety-induced observations and evaluate the influence. For example, will subjects w/o anxiety behave differently (when the differences can be detected) or if it will have greater influence among subjects w/ anxiety. In addition, it seems that data used for analyses were collected during just one collection session. Could the author explain why the subjects were not asked to perform multiple sessions and use the averaged data instead? What if a subject walked a bit differently because he/she was nervous, under the influence of adrenalin with faster heart rate, or just some random effect?

Thirdly, after reading the Results section, the reviewer noticed that most of these statistically significant differences were not material, within 2 or 3 degrees of angle, which raised the concerns over the biological plausibility. First of all, the audience should understand the pros and cons of an IMU-based motion capture system. With general design features, a typical IMU system can detect angular change greater than 1 or 2 degrees of angle. According to the company, the Opal sensor has dynamic accuracy of 2.8 degrees. Therefore, unlike the evaluations of other large joint movements, with differences so close to the margin of error, it may not be so straightforward to interpret the results.

Aforementioned are some of the major concerns over the quality and readiness of this manuscript.

Some specific questions/suggestions,

Abstract, “5 top features at identifying anxious individuals …” only included 4 in the parenthesis

Page1, line6, missing “than” between “more” and “three”

Page1, line8-9, redundant usage of “annual”

Page2, line13, should be “individual’s”

Page2, line14, should be “use” instead of “using”

Page2, line22-23, this statement seems detached from the prior literature review, where it states “these findings suggest the need to …. using objective measures”. To serve the purpose of this study, gait and balance may be objective measures; but they are also indirect ones if they are chosen to evaluate someone’s anxiety level. Maybe the authors should provide greater details on how self-reported measures are complemented by indirect and object measures.

Page2, line25-26, the wording needs revision

Page2, line31-32, missing “they” between “because” and “suggest”

Page2. Line34, should be “Zhao”

Page3, line85, delete “I” between “has to be” and “between”

Page3, line86-91, the wording needs revision. “Individuals were excluded … were included in this study”

Page3, line91, 133 “were” qualified

Page4, line112, should be “range”

Page5, line166-170, please consider presenting the formulas in dedicated open space

Page5,6,7, there may be too much redundancy in wording when referring to the two group of subjects, please consider an easier and more concise way to communicate and save room on wording counts for additional information or discussion.

At this moment, the reviewer decided to bring up major concerns with a few examples of spelling errors and grammar issues. The reviewer believes that the comments provided in this report should be addressed or explained by the authors first before moving forward with additional review. Please take your time to prepare your response to clarify all the issues, concerns, and misunderstandings, the process would only help strengthen your work.

Thank you very much!

Author Response

Dear Reviewer,

We would like to thank the reviewer for their time and excellent feedback. We feel that based on the reviewer’s feedback, our manuscript is significantly improved. Please see the reviewer’s comments and responses below in red.

Reviewer 2

This is a review of the manuscript “Identifying individuals who currently report feelings of anxiety using walking gait and quiet balance: An exploratory study using machine learning”, submitted to the “Intelligent Sensors” section, special issue “Analysis of Biomedical Signals and Physical Behavior Sensing in the Development of Systems for Monitoring, Training, Controlling, and Improving Quality of Life”.

This manuscript describes a pilot study using machine learning (ML) to simultaneously monitor individuals’ gait and quiet balance to potentially identify individuals under the influence of anxiety. Although the concept may be interesting to the readers of Sensors and this special issue, especially among those clinicians who are looking for an objective and reliable surveillance and/or screening tool for psychological assessment on anxiety. However, the current manuscript was not written with clear logic and hard to follow with the presence of spelling errors, grammar issues, and redundant wordings. It could use some extensive editing work to make the main text fluent and easier to understand. More importantly, after reading the manuscript, the reviewer developed some concerns over the quality of this manuscript in its current format, particularly the study design, the way data were analyzed, and the interpretation of results. Some additional explanation would be good.

We appreciate the reviewer pointing out the grammatical and spelling errors. We have had multiple authors proofread the manuscript to correct for these.

In general, based on the overall impression of the current exploratory study, the reviewer would say that the authors aimed to establish an objective approach to identify/predict feelings of anxiety using machine learning with a pool of candidate variables of the gait and balance features, as stated in the Introduction section, “to identify whether balance control, gait, or a combination of the two would be most accurate to predict if a healthy individual is feeling anxious in the moment”. To the reviewer’s knowledge, gait parameters, balance control, etc. are closely related to neuromuscular functions, which can be affected by the psychological status and/or disturbance, not necessarily limited to or caused by anxiety alone. Therefore, it appeared to be a long leap trying to establish a direct connection using gait and balance as the outcome measure of anxiety. The current Introduction section is not focused on providing the foundation work, such as the existing evidence of the associations between gait parameters and postural control capability (i.e., balance test) and the psychological features (i.e., anxiety), especially some concrete evidence explaining the biological plausibility, in what particular way anxiety would actually influence how an individual walk or stand. If the logic is not offered or clearly presented, the whole study might become a pursue of pure statistics, just throwing a whole bunch of variables out there and let the algorithm run and see what kind of relationship will show up. In the Discussion section, the authors do provide some insight when discussing Objective 1 and 3. The reviewer would suggest adding similar material in the Introduction. Most importantly, the authors would need to explain the causal relationship that may warrant a predictive relationship between how people feel and how they walk or balance. It is one thing that they are statistically associated and another thing if you can rely on one to predict another. In practice, if indeed an individual is exhibiting large deviation from the “normal” walking gait, we do not immediately seek explanations or diagnosis from a psychology perspective. Instead, clinicians will more likely look for neuromuscular and/or skeletal issues first. Therefore, what is the relevance of these findings to clinical practices?

We appreciate the reviewer’s feedback. We have now added potential explanations as to why gait and balance might be different in individuals who experience feelings of anxiety. Although this work is not aimed at diagnosing clinical anxiety using machine learning (significant work already exists in that realm), the objective of this study was to determine whether the authors could identify individuals who are feeling anxious in the moment. As we move towards personalized medicine the first part of being able to prescribe individualized interventions is to identify various conditions (i.e. depression, anxiety, fatigue, etc). With deviations as small as what we see in our sample of healthy adults we do not believe clinicians would be able diagnose anxiety, nor should they base their diagnoses on the findings of our study. We present these findings as a way to show how technology can be used to identify these moods through human movement. We hope that this work is the beginning of bridging the gap between computer scientists, psychologists and clinicians to help us create technologies that may be used to identify mood states in healthy individuals using human movement. We also hope that clinicians such as physiotherapists who do perform gait-based assessments may seek to understand that potential variations in gait may also be associated with feelings of anxiety.

Secondly, as the authors mentioned in the Limitations section that this is a cross-sectional design with self-reported anxiety status collected from a relatively small sample of young subjects, who were separated into two groups, one with anxiety (≥3) and the other without anxiety (=0), the current methodology may not just have limitations but flaws in logic. The current comparisons were made between datasets collected from different subjects rather than from the same subject at various status (i.e., use his/herself as reference for what is normal and what is abnormal). Unfortunately, with the vast variations in human neuromuscular-skeletal system, it is not an easy job to establish a standard that is true and applicable to all individuals, not to mention the influence of other variables, including gender, age, body composition, general health, and medical conditions. The current sample while being young and healthy, is also a substantial limiting factor on the possible application of the current findings among other samples, where subjects do have this or that condition, such as obesity, sleep problems, degenerative changes to the neuro-musculoskeletal system. With everything blended in together, how can we single out the influence of anxiety? As mentioned by the authors, since anxiety-inducing conditions can be simulated (16,19), the current study may consider establishing baseline and anxiety-induced observations and evaluate the influence. For example, will subjects w/o anxiety behave differently (when the differences can be detected) or if it will have greater influence among subjects w/ anxiety. In addition, it seems that data used for analyses were collected during just one collection session. Could the author explain why the subjects were not asked to perform multiple sessions and use the averaged data instead? What if a subject walked a bit differently because he/she was nervous, under the influence of adrenalin with faster heart rate, or just some random effect?

We appreciate the reviewer’s concerns. While we agree that creating anxiety inducing conditions may have been ideal, we were trying to stay as close to the methodology used by Zhao and colleagues (2019) and Miao and colleagues (2021). In an effort to remain as consistent as possible with the current literature these authors used the same methodology. These studies (Zhao, et al, 2019 and Miao, et al, 2021) used a single session, a 2-minute walk around a 6m track and tried to identify individuals who felt anxious using machine learning, each of these factors were emulated in our study. However, our goals were slightly different in that we wanted to be able to identify individuals who were feeling anxious in the moment rather than those had reported feeling anxious over the last 2 weeks (Zhao, et al, 2019, Miao et al, 2021). Further we wanted to examine variables most commonly used in gait literature instead of the way gait was measured in those two studies (i.e. movement in left ankle, without any explanation of how the ankle movement was different, or movement of the ear and nose). Further, the work by Zhao, et al, 2019 and Miao, et al, 2021 was conducted using X-Box Kinect cameras which are data intensive. The goal of this study was to determine whether we could replicate their findings without using data intensive methods. Therefore, we chose to use IMUs that provide us with useful gait measures commonly reported in literature, while having low data outputs.

Another reason why we chose not to induce feelings of anxiety in our participants is that previously published work suggests that not everyone responds to anxiety inducing conditions (i.e. some individuals may not feel anxious at all when trying to induce anxiety). We have cited one of those published manuscripts (Fuller, et al, 2021). We have also added this to our limitations section.

Therefore, considering the exploratory nature of this study, these are the many reasons why we chose to replicate the methodologies published by Zhao, et al, (2019) and Miao, et al, (2021) and we chose not to create an intervention to induce feelings of anxiety. However, based on these cross-sectional findings we hope to create anxiety provoking situations to determine whether we can identify when someone becomes anxious compared to when they are not. Further, with what we know about inter-individual differences in responses to various anxiety provoking conditions (Fuller, et al, 2021), we also hope to use this work to also identify individuals who do not report a change in feelings of anxiety after an anxiety provoking interventions..

Zhao N, Zhang Z, Wang Y, Wang J, Li B, Zhu T, et al. See your mental state from your walk: Recognizing anxiety and depression through Kinect-recorded gait data. PLoS one. 2019;14(5):e0216591.

Miao B, Liu X, Zhu T. Automatic mental health identification method based on natural gait pattern. PsyCh Journal. 2021;

Fuller DT, Smith ML, Boolani A. Trait Energy and Fatigue Modify the Effects of Caffeine on Mood, Cognitive and Fine-Motor Task Performance: A Post-Hoc Study. Nutrients. 2021;13(2):412

Thirdly, after reading the Results section, the reviewer noticed that most of these statistically significant differences were not material, within 2 or 3 degrees of angle, which raised the concerns over the biological plausibility. First of all, the audience should understand the pros and cons of an IMU-based motion capture system. With general design features, a typical IMU system can detect angular change greater than 1 or 2 degrees of angle. According to the company, the Opal sensor has dynamic accuracy of 2.8 degrees. Therefore, unlike the evaluations of other large joint movements, with differences so close to the margin of error, it may not be so straightforward to interpret the results.

We agree with the reviewer’s concerns about 2-3 degrees angle differences may be statistically significant but not clinically meaningful. However, the objective of this study was not to provide clinically meaningful findings, instead it was to use gait features that have clinical meaning (i.e. gait speed instead of left foot movement as used by previous authors). We understand the concern about the Opal monitors having a 2.8 degrees dynamic accuracy which may impact the smaller movement differences. However, our goal was to identify which joints were statistically different so that researchers interested in this work may use our findings as a guide for their work, instead of guessing how gait might be different. Further, we ran our models 10,000 times using a Monte Carlo method by removing 10% of our total subjects randomly from our data with each iteration. This validation method allows us to statistically account for the small dynamic accuracy differences in the Opal monitors. The Monte Carlo method for cross validation of data is best for sensor-based data, as it allows us to account for data that may have dynamic accuracy issues such as what the reviewer brought up regarding the the Opal sensors (Girard, 1989, Metropolis, et al, 1949, Xu, et al, 2001). We have now cited this in our statistical analysis section.

Girard A. A fast ‘Monte-Carlo cross-validation’procedure for large least squares problems with noisy data. Numerische Mathematik. 1989;56(1):1–23.

Metropolis N, Ulam S. The monte carlo method. Journal of the American statistical association. 1949;44(247):335–41.

Xu Q-S, Liang Y-Z. Monte Carlo cross validation. Chemometrics and Intelligent Laboratory Systems. 2001;56(1):1–11.

Aforementioned are some of the major concerns over the quality and readiness of this manuscript.

Some specific questions/suggestions,

Abstract, “5 top features at identifying anxious individuals …” only included 4 in the parenthesis

We appreciate the reviewer pointing this out. We can understand how this was confusing. We lumped lumbar rotation and lumbar movement in the sagittal plane into one (lumbar movement). We have now corrected this.

Page1, line6, missing “than” between “more” and “three”

Thank you. It has been corrected

Page1, line8-9, redundant usage of “annual”

Thank you. It has been corrected

Page2, line13, should be “individual’s”

Thank you. It has been corrected.

Page2, line14, should be “use” instead of “using”

Thank you. It has been corrected.

Page2, line22-23, this statement seems detached from the prior literature review, where it states “these findings suggest the need to …. using objective measures”. To serve the purpose of this study, gait and balance may be objective measures; but they are also indirect ones if they are chosen to evaluate someone’s anxiety level. Maybe the authors should provide greater details on how self-reported measures are complemented by indirect and object measures.

We have now corrected this sentence to state that we must find objective direct and indirect measures such as biomarkers and walking gait or balance, which may be impacted by feelings of anxiety.

Page2, line25-26, the wording needs revision

Thank you. We have corrected this sentence.

Page2, line31-32, missing “they” between “because” and “suggest”

Thank you. It has been corrected.

Page2. Line34, should be “Zhao”

Thank you. It has been corrected.

Page3, line85, delete “I” between “has to be” and “between”

Thank you. It has been removed.

Page3, line86-91, the wording needs revision. “Individuals were excluded … were included in this study”

Thank you. We have now removed “were included in this study”

Page3, line91, 133 “were” qualified

We have added “were”

Page4, line112, should be “range”

Thank you. We have now corrected this.

Page5, line166-170, please consider presenting the formulas in dedicated open space

We have now reported all formulas used in this study to calculate gait parameters.

Page5,6,7, there may be too much redundancy in wording when referring to the two group of subjects, please consider an easier and more concise way to communicate and save room on wording counts for additional information or discussion.

We have now read through the entire discussion and tried to reduce redundancy.

At this moment, the reviewer decided to bring up major concerns with a few examples of spelling errors and grammar issues. The reviewer believes that the comments provided in this report should be addressed or explained by the authors first before moving forward with additional review. Please take your time to prepare your response to clarify all the issues, concerns, and misunderstandings, the process would only help strengthen your work.

We’d like to thank the reviewer for these substantial comments.

Thank you very much!

We’d like to thank the reviewer for this extensive review. We appreciate the effort in helping us improve this manuscript.

Reviewer 3 Report

The study aims at defining through machine learning algorithms a series of indicators obtained from walking and from balance tests able to identify the state of anxiety. It is very interesting and overall well conducted. However, I have some suggestions / queries.

Methods:

Methods:

The authors use a young population 18-36 as inclusion criteria, please explain why it was chosen to use only a young population.

Furthermore, the study reports in the results and discussion section a series of rather limited variations in the numerous variables identified, I wonder how the authors have contained any confounding factors. It is true that subjects with pathologies and who manifested pain that could interfere with the biomechanics of walking were excluded, but some variables such as limb oscillation can also be influenced by minimal postural alterations that do not necessarily cause pain but can cause a modification of gait biomechanics. I ask the authors to clarify this at least in the limitation of the study.

Results.

I would report the 21 significant features in the text or in a specific table inserted in the text rather than only in the supplementary file.

Discussion.

I would modify the discussion section by removing the subtitles Objective 1 Objective 2 Objective 3 and possibly recalling the different objectives in the text in a discursive way.

Small fixes:

page 3 line 85: there is an "I" which I believe should be removed

page 3 line 105: replace mods with moods.

Page 4 line 112: replace rnage with range

Page 6 line 217: Enter the RF acronym for random forest here and correct on line 222

Author Response

Dear Reviewer,

We would like to thank the reviewer for their time and excellent feedback. We feel that based on the reviewer’s feedback, our manuscript is significantly improved. Please see the reviewer’s comments and responses below in red.

Reviewer 3

The study aims at defining through machine learning algorithms a series of indicators obtained from walking and from balance tests able to identify the state of anxiety. It is very interesting and overall well conducted. However, I have some suggestions / queries.

Methods:

Methods:

The authors use a young population 18-36 as inclusion criteria, please explain why it was chosen to use only a young population.

We appreciate the reviewer’s concern in using a healthy population between the ages of 18-36. The goal of this study was to replicate the work by Zhao and colleagues (2019) and Miao and colleagues (2021) as closely as possible while using gait parameters most commonly used in literature.

Furthermore, the study reports in the results and discussion section a series of rather limited variations in the numerous variables identified, I wonder how the authors have contained any confounding factors. It is true that subjects with pathologies and who manifested pain that could interfere with the biomechanics of walking were excluded, but some variables such as limb oscillation can also be influenced by minimal postural alterations that do not necessarily cause pain but can cause a modification of gait biomechanics. I ask the authors to clarify this at least in the limitation of the study.

Thank you for this feedback. We have now clarified this in the limitations section.

Results.

I would report the 21 significant features in the text or in a specific table inserted in the text rather than only in the supplementary file.

Thank you for this suggestion. We have now provided a table that can be inserted into the text.

Discussion.

I would modify the discussion section by removing the subtitles Objective 1 Objective 2 Objective 3 and possibly recalling the different objectives in the text in a discursive way.

Thank you for this suggestion. We have now followed the reviewer’s suggestion and removed the subtitles and discussed our findings in a discursive way.

Small fixes:

page 3 line 85: there is an "I" which I believe should be removed

Thank you for pointing this out. We have corrected it.

page 3 line 105: replace mods with moods.

Thank you for pointing this out. We have corrected this.

Page 4 line 112: replace rnage with range

Thank you. We have now corrected this.

Page 6 line 217: Enter the RF acronym for random forest here and correct on line 222

Thank you for this feedback. We have addressed this concern.

Round 2

Reviewer 2 Report

The authors have sufficiently addressed all my suggestions and comments. Thank you for the revision and detailed explanation.